# Multi-Channel Spectral Sensors as Plant Reflectance Measuring Devices—Toward the Usability of Spectral Sensors for Phenotyping of Sweet Basil (*Ocimum basilicum*)

Trung Nam Tran [1], Rieke Keller [1], Vinh Quang Trinh [2,*], Khanh Quoc Tran [2] and Ralf Kaldenhoff [1]

[1] Department of Biology Applied Plant Sciences, Technical University of Darmstadt, Schnittspahnstr. 10, 64287 Darmstadt, Germany; tran@bio.tu-darmstadt.de (T.N.T.); rieke.keller@stud.tu-darmstadt.de (R.K.); kaldenhoff@bio.tu-darmstadt.de (R.K.)

[2] Laboratory of Adaptive Lighting Systems and Visual Processing, Technical University of Darmstadt, Hochschulstr. 4a, 64289 Darmstadt, Germany; khanh@lichttechnik.tu-darmstadt.de

* Correspondence: vinh@lichttechnik.tu-darmstadt.de; Tel.: +49-6151-16-22881

**Abstract:** Modern agriculture demands for comprehensive information about the plants themselves. Conventional chemistry-based analytical methods—due to their low throughput and high associated costs—are no longer capable of providing these data. In recent years, remote reflectance-based characterisation has become one of the most promising solutions for rapid assessments of plant attributes. However, in many cases, expensive equipment is required because accurate quantifications need assessments of the full reflectance spectrum. In this experimental study, we examined the versatility of visible spectral sensors as alternative reflectance measuring devices for biological/biochemical quantifications of sweet basil (*Ocimum basilicum*). Our results confirm the applicability and scope of visible spectral sensors for analysis and quantification of important plant properties, in particular the contents of valuable substances, such as phenolic compounds and flavonoids.

**Keywords:** reflectance; *Ocimum basilicum*; colour sensor; phenotyping

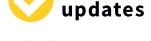



## 1. Introduction

The emergence of modern precision agriculture (so-called Agriculture 4.0) can be described as a fortuitous resonance of several technical and scientific advancements [1]. The complete sequencing of many plant genomes [2] and the rapid advances in genetic and metabolic engineering [3] are leading to a deeper understanding of many biological and biochemical processes in plants and allow extensive manipulations of plants' metabolic networks. Soil-independent agricultural techniques such as hydroponics, aquaponics and aeroponics are shifting plant production from crop field to indoor plant factories. Here, a wide range of modern technologies control plant cultivation [4]. Automation [5], sensors [6], unmanned aerial vehicles (*UAV*) [7] and telecommunication techniques [8] enable a round-the-clock, real-time, multi-faceted monitoring and controlling of the production process. The leaping improvements in data science, artificial intelligence (*AI*) [9], computer processing power and the speed and capacity of Internet connectivity [10] enable the creation of sophisticated decision-making algorithms with minimal human intervention and unprecedented speed and precision that were just fantasy a few decades ago. These technologies demand high quality and real time data. Thus, it is desirable to collect similar powerful methods to collect plant-related data for crop quality assessments.

A variety of information can be achieved by analytical chemistry and molecular biology, such as taste, flavours, ripeness, nutrient contents and minor concentrations of numerous active compounds. While the methods themselves are very accurate and reliable, many are comparable slow, are destructive, have low-throughput and are high-cost both in terms of time and money. In addition, these often require well-equipped laboratories

and skilful technicians. These drawbacks are increasingly evident considering the changes in agriculture in the first decades of the 21st century. Recently, new approaches for the rapid quantification of plant biology and biochemistry parameter were established [11]. The method described herein employs the principle of reflectance-based measurements. When light reaches the surface of the plant, it can be absorbed by plant tissues (absorbance), transmitted through the plant and emerge on the other side (transmittance) or be reflected (reflectance). The degree of reflection depends on many factors, including the light's wavelength, the angle of incidence and importantly, the optical characteristics of plant tissues, which are in turn determined by its structural, biological and biochemical properties (e.g., plant structure, surface roughness, tissue thickness and density and pigment contents) [12]. Therefore, it should be possible to receive information about plants from reflectance. The endeavour to elucidate this relationship has given rise to several new interdisciplinary research areas that were developed in recent years: remote sensing, hyperspectral imaging, optical contactless measurements or chemometrics [13].

Reflectance spectra of green leaf materials typically show low reflectance in blue and red regions, high in green wavelengths and high in near-infrared (*NIR*) wavelengths [14]. Such features are attributed to the occurrence of two major plant pigments: chlorophylls and carotenoids. The concentrations and ratios of these pigments serve as direct indicators of plant status. For example, low chlorophyll content indicates nitrogen deficiency, and a low chlorophyll/carotenoid ratio points to advanced senescence status. The first generation of reflectance-based plant analysis methods relies on a small number of wavelengths, from which information regarding the abundance of chlorophylls and carotenoids was estimated. The simplicity and robustness of this approach supports its popularity across a wide range of conditions [15]. The necessary sensors are readily available as relatively low cost investments. Since the relationship between plant status and its pigment composition is established, interpretation of data is straightforward, on the one hand. On the other hand, only a limited number of plant traits can be characterised from pigment information only.

Most plant phenotypes do not correlate directly with reflectance. For example, many secondary metabolites are colourlessness and occur at low concentrations. They give small, almost invisible reflectance footprints. Their production is generally independent from chlorophyll and carotenoid biosynthesis; thus, there is no direct correlation with pigment contents. Their relationships with reflectance are therefore indirect, having many intermediate factors. To predict such phenotypes, it is necessary to treat the connection between the complete reflectance spectra and the corresponding plant phenotype as a black box and apply the principles of machine learning (*ML*) to build prediction models for the latter based on the former [16]. The measured reflectance spectra range between the visible (350–780 nm) and *NIR* regions (780–1100 nm) and often extend to shortwave infrared wavelengths (1100–2500 nm). The major advantage of this approach is that a priori knowledge regarding the relationship between reflectance and biological information is not required; therefore, it is supposedly applicable for a great number of plant traits [17–19]. As examples, the fat content of peanuts [20], the sugar contents of kiwifruit [21], the lycopene and phenolic content of tomatoes [22], the firmness of wheat [23], the fungal infection status of peanuts [24], early rottenness in apples [25] and viral infections of watermelon [26] can all be reliably predicted. Su and Sun (2018) provided a comprehensive list of applications of reflectance spectrum analysis [14]. However, this approach is not without caveats because equipment for acquisitions of full spectra, such as spectrophotometer or hyperspectral camera, are expensive. Their cost and size were significantly reduced in recent years but are still high enough to hold up widespread applications. For a widely used modern agriculture technology, it should be sensible to combine the positive features of the approaches: the utilisation of *ML* algorithms with reflectance data collected from low-cost optical sensors. The advancement of sensor technology in recent years, which resulted in a new generation of inexpensive, compact, yet very powerful spectral sensors, made applications in agriculture possible.

A photodiode is the central element of a spectral sensor. It consists of a semiconductor p–n junction device to convert an incident light-photon to electrical current. In general, this current is very low, in the order of micro or nano amperes, and should be converted and amplified by an operational amplifier (*OPAM*). Subsequently, the analogue signal is converted into a digital form by an analogue–digital converter (*ADC*). Besides these core and signal processing elements, optical filters are another essential element of spectral sensors. Singular colour sensor channels—such as red, green, or blue filter sensors detect the corresponding wavelengths of incident light. Simple singular colour sensors were generally utilised for monitoring the brightness levels of a specific colour. As of the last few years, several specific wavelength channel filters can be combined for a more complex type of multi-channel spectral sensors. Currently, very sophisticated spectral sensors with higher spectral wavelength channels, such as *AS7341* (eight visible optical channels, three extra sub-channels for measuring *NIR* and checking full incident light spectra and their status, size 3.1 × 2 × 1, 1.8 V, operating from −30 to 70 °C), *AS7343* (14 visible and *IR* channels; see Figure 1) and *AS7265x* (18 visible and *NIR* channels from 410 to 940 nm, 16-bit *ADC*s) are commercially available [27–30]. They also can be integrated into cameras through the use of Bayer filter arrangements and image sensors, such as *CCD* or *CMOS* [31]. In recent years, the cost of colour sensors was reduced significantly, allowing a much wider range of applications including those in agriculture and horticulture [30]. Several horticultural applications of spectral sensors have also been reported in the literature: assessments of plants' nutritional and physiological status [32,33], chlorophyll determination [34], monitoring of plant growth [35] and weed recognition [36].

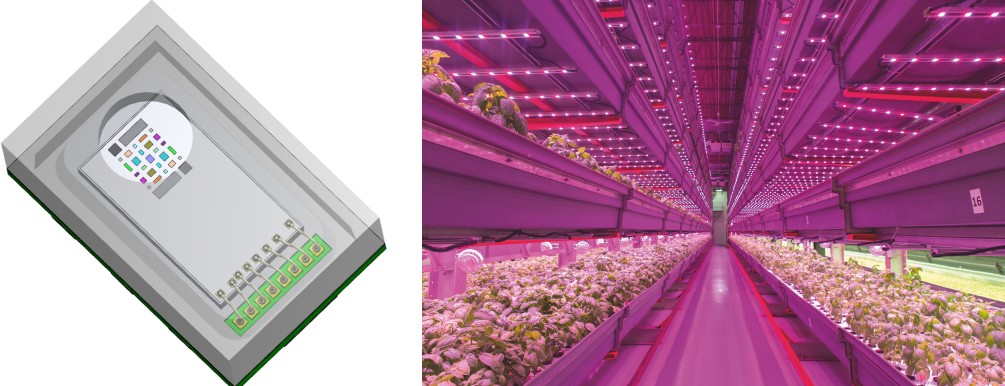

**Figure 1.** *AS7343* provides a fast and accurate spectral measurement of incident light for crop cultivation (source: ams Osram [30]).

Far greater potential for the application of such spectral sensors in fully automated vertical farming facilities can be predicted. Effective contactless monitoring and controlling of plant status is feasible, and the results are comparable to those from hyperspectral imaging; the former is less expensive though. With such systems, the growth, development and health status of plants and their interactions with environmental conditions (temperature, humidity, light, water and fertiliser) can be accurately and continuously measured, digitally monitored and optimised. For practical applications, the performances of visible spectral sensors regarding stability, accurateness and reliability are superior to those of far-infrared (*FIR*), *NIR* and ultraviolet (*UV*) sensors. These sensors often suffer from high signal to noise ratios, degradation or input value shifting.

This work aims to provide an experimental study on the applicability of spectral sensors to describing various biological and biochemical properties of plants. In order to simulate the action of a colour sensor array, the following restrictions were set:

- Instead of a hyperspectral imaging camera, the experimenters collected reflectance data by taking many point measurements of the leaf upper side (adaxial) reflectance. This simulates the action of a colour sensor array.

- Unlike hyperspectral imaging cameras, spectral sensors do not yield spatial information and cannot distinguish between plant substructures. Therefore, within one plant we could not distinguish between leaves with different leaf ages or leaf positions.
- Reflectance was measured in the visible wavelength range, i.e., from 380 to 800 nm, which is comparable to visible sensors in practical applications.
- Plants of different ages and from different light conditions were analysed, reflecting the natural variations in plant-characteristics under these conditions. Three light conditions were chosen: white, red and blue/red. These colours are widely used in indoor agricultural and horticultural applications.

The authors of this paper intended to answer the following questions:

- The aforementioned restrictions will have a negative impact on the predictive power of the acquired reflectance data. Which biological/biochemical characteristics could still be observed reliably?
- For such parameters to be predicted, the measured spectral reflectances must be converted into the predicted values using so-called spectral weighting functions. Section 2.9 contains a detailed mathematical description of the spectral weighting function. The accuracy and robustness of the spectral weighting functions are critical for prediction accuracy. What are the defining features of spectral weighting functions?
- When full spectra are acquired with a spectrophotometer or hyperspectral camera, the spectral weighting functions are processed using very simple mathematical calculations. Is it possible for multi-channel spectral sensors to process such weighting functions? If not, how can they be modified to perform this task?

As a model, we chose sweet basil (*Ocimum basilicum*), a well-known herb used by humans for centuries. The basil system has several advantages: fast plant growth and simple cultivation; the plant is rich in secondary metabolic compounds, whose concentrations vary depending on age and growth conditions. Six plant parameters were chosen for analysis: specific leaf weight ($SLW$); total phenolic content ($TPC$); total flavonoid content ($TFC$); and the concentrations of the main pigments: chlorophyll a ($ChA$), chlorophyll b ($ChB$) and total carotenoids ($CaT$).

## 2. Materials and Methods

### 2.1. Overview of the Experimental Plan

In total, 27 basil plants were used for the study. These were divided into three groups of nine plants each, three weeks after sowing. Each group was cultivated under three different light conditions, white light, red light and mixed blue/red light, for another three weeks. At 5, 14 and 21 days after the onset of the experiments, three plants from each group were taken for analysis (Figures 2 and 3).

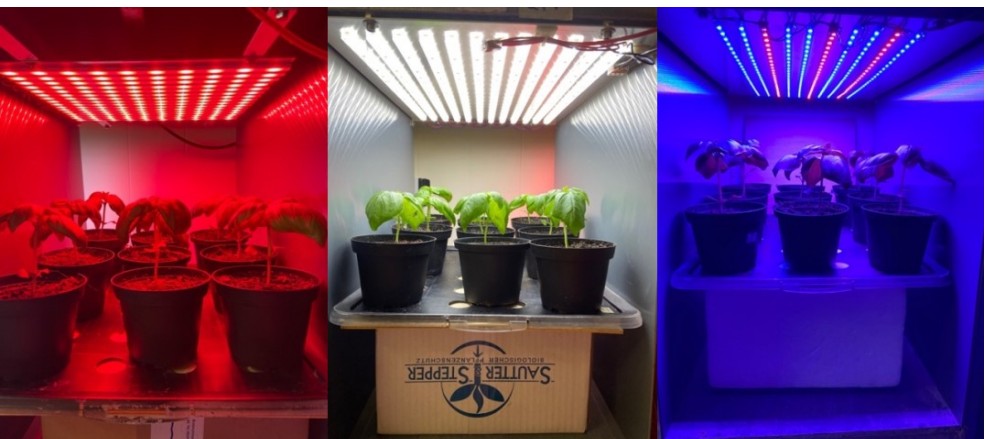

**Figure 2.** Cultivation of sweet basil under three different light conditions.

From each plant, ten leaf discs were excised for the determination of *SLW* and for reflectance measurements. The latter was performed on each leaf disc, and the average of ten measurements was considered as representative for the whole plant. The rest of the leaf material was frozen in liquid nitrogen and treated for extraction of secondary-coloured compounds.

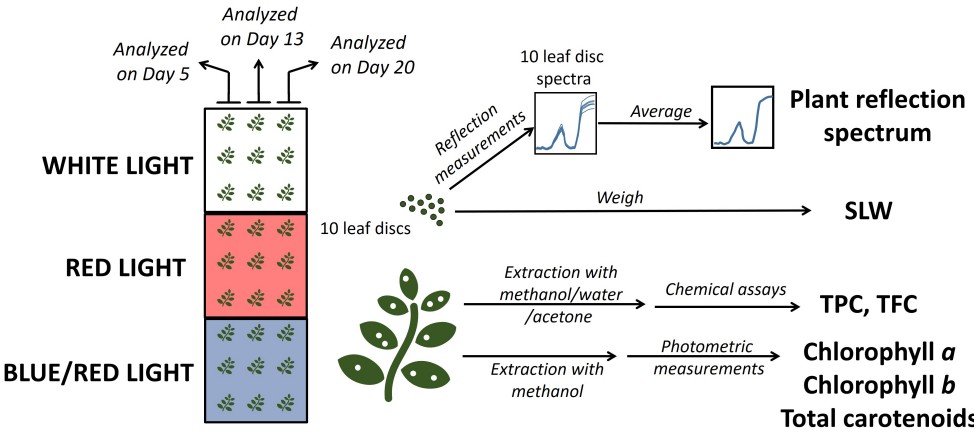

**Figure 3.** Overview of the experimental plan for determination of reflectance spectrum and biological and biochemical properties. *SLW*—specific Leaf Weight; *TPC*—total phenolic Content; *TFC*—total flavonoid content.

## 2.2. Plant Cultivation

*Ocimum basilicum* (cultivar Genovese) was cultivated in soil pots (Fruhstorfer Erde Typ *T*, Hawita) under greenhouse conditions. Temperature was maintained between 19 and 23 °C with relative humidity of 50–60%. Three weeks after sowing, 27 young plants were transferred to a phytochamber (temperature 20 °C, relative humidity 50%). There, they were divided into three groups of nine plants each. These groups were further cultivated under three different light conditions: white, red and a blue/red light combination (3/1). The day/night cycle was set to 16 h light:8 h dark. Light intensity measured at the upper leaves was 100 μmol/m² · s. The lights were adjusted daily to keep the light intensity constant as the plants grew. The spectra of the three LED light conditions used in the experiments are shown in Figure 4.

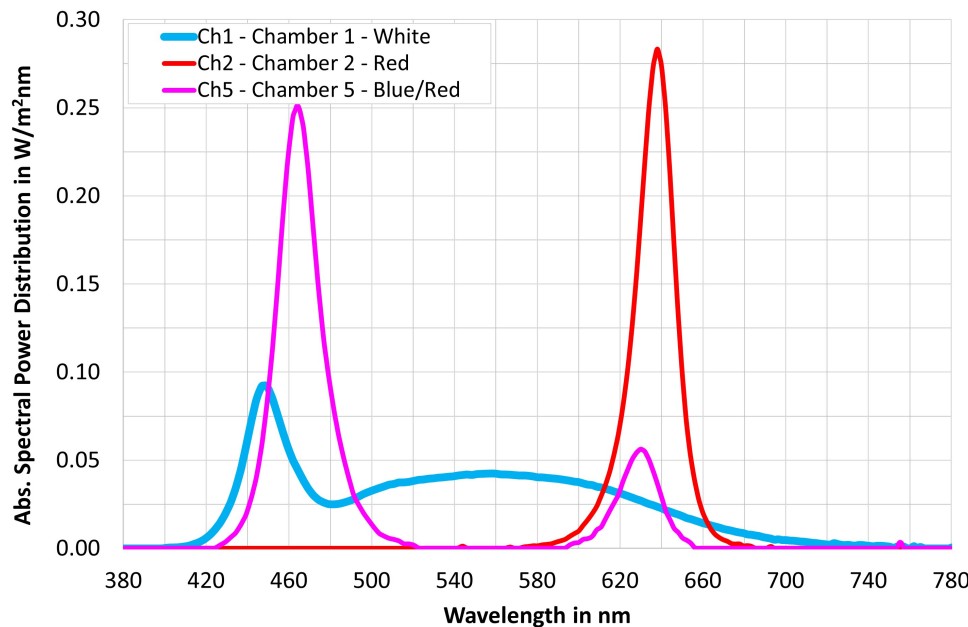

**Figure 4.** Absolute spectral power distributions of the three light conditions used in the experiments.

### 2.3. Analysis of Plants' Parameters

Determination of specific leaf weight (*SLW*): From each plant, ten leaf discs (0.95 cm in diameter) were randomly excised and immediately placed on a pre-weighed water agar plate (1%) to avoid desiccation (Figure 5). The agar plate weight was measured again and the difference ($\Delta W$) estimated. *SLW* can be calculated with the following formula:

$$SLW \left[\frac{g}{cm^2}\right] = \frac{\text{Leaf weight}}{\text{Leaf surface}} = \frac{4 \cdot \Delta W}{10 \cdot \pi \cdot 0.95^2} \tag{1}$$

Within 10 min after excision, the leaf discs on the agar plate were used for reflectance measurements.

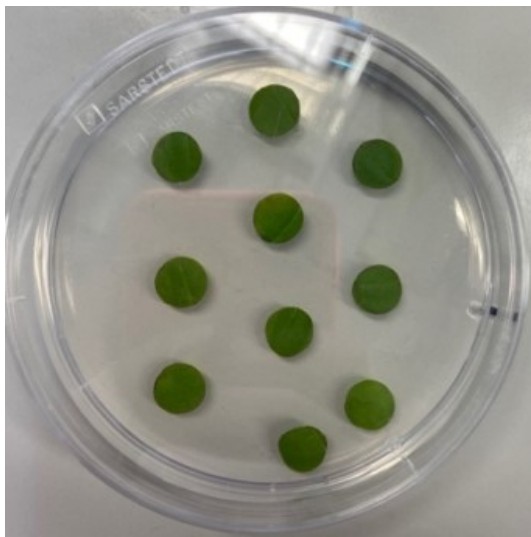

**Figure 5.** Leaf discs on an agar plate.

### 2.4. Plant Extracts

The remaining leaf material was frozen in liquid nitrogen and homogenised. Leaf powder weight was assessed. For extraction of phenolic compounds and flavonoids, 1 mL of methanol/water/acetone (60/30/10 *v/v/v*, freshly prepared) was added to each 250 mg of leaf material. The suspension was mixed vigorously and incubated at 4 °C for 15 min. Subsequently, cell debris was removed by centrifugation at 4 °C and 13,000 rpm for 10 min. The clear supernatant was transferred to a fresh tube. The extraction was repeated once more, and the supernatants were combined.

For extraction of photosynthetic pigments, 10 mL methanol was pre-treated with 150 mg calcium carbonate. To each 150 mg of frozen leaf powder, 1 mL pre-treated methanol was added, and the mixture was centrifuged at 4 °C and 13,000 rpm for 10 min. The extraction was repeated twice, and the supernatants were combined.

### 2.5. Quantification of Total Phenolic Contents (*TPC*)

*TPC* of the methanol/water/acetone extract was determined with the Folin–Ciocalteu assay [37]. A calibration curve was created with standard solutions of gallic acid ranging from 250 to 750 mg/L. The total amount of phenolic content in the extract was determined from comparison to the calibration curve. *TPC* was calculated as the total amount of phenolic content per fresh weight unit.

$$TPC \left[\frac{\text{mg gallic acid equivalent}}{\text{g fresh weight}}\right] = \frac{\text{Total amount of phenolic compounds in extract}}{\text{Fresh weight used for extraction}} \tag{2}$$

### 2.6. Quantification of Total Flavonoid Contents (TFC)

*TFC* of the methanol/water/acetone extract was determined by colorimetric measurement according to Zhishen et al. [38]. For calibration, solutions of (+)-catechin with concentrations ranging from 100 to 250 mg/L were used. *TFC* was calculated as the total amount of flavonoids per fresh weight unit.

$$TPC \left[\frac{\text{mg catechin equivalent}}{\text{g fresh weight}}\right] = \frac{\text{Total amount of flavonoids in extract}}{\text{Fresh weight used for extraction}} \tag{3}$$

### 2.7. Quantification of Chlorophyll a, Chlorophyll b and Total Carotenoids

The concentrations of chlorophyll a (*ChA*), chlorophyll b (*ChB*) and total carotenoids (*CaT*) were determined from the absorbance at 470, 653 and 666 nm [39] using the following equations:

$$ChA\ [\text{mg/L}] = 15.65 \cdot A_{666nm} - 7.34 \cdot A_{653nm} \tag{4}$$

$$ChB\ [\text{mg/L}] = 27.05 \cdot A_{653nm} - 11.21 \cdot A_{666nm} \tag{5}$$

$$CaT\ [\text{mg/L}] = \frac{1000 \cdot A_{470nm} - 2.86 \cdot [\text{Chlorophyll a}] - 129.2 \cdot [\text{Chlorophyll b}]}{245} \tag{6}$$

Pigment content was calculated by dividing the pigment concentration in the extract by the fresh weight of plant material used for extraction.

### 2.8. Spectral Reflectance—Measurement Systems and Spectral Features

The measurement of leaf reflectance (Figure 6) requires a light source (*LS*), spectral camera (*CS*), white standard (*WST*) and leaves (*L*).

The task of the light source *LS* is to generate a stable spectrum $S(\lambda)$ that does not change regardless of the operating temperature or burning time. A tungsten halogen lamp with a continuous spectrum in the visible range was applied during the measurement. Prior to data assessment, the lamp was switched on for 1 hour and the spectrum stability was verified.

The standard spectral reflectance $WST(\lambda)$ of the white light standard, which consists of the $BaSO_4$ material, is approximately 1.0 across the full visible wavelength range (from 380 to 780 nm) according to the calibration document of the manufacturer.

The spectroradiometer CS2000 (CS) from Konica Minolta was used for spectral reflectance measurements (the manufacturer's information is given in more detail in [40]). Initially, it measured the spectral radiance $S(\lambda) \cdot WST(\lambda)$ (Figure 6, left). Subsequently, the light source spectrum could be determined by Equation (7).

$$S(\lambda) = \frac{S(\lambda) \cdot WST(\lambda) \text{ measured by CS}}{WST(\lambda) \text{ supplied by the manufacturer}} \tag{7}$$

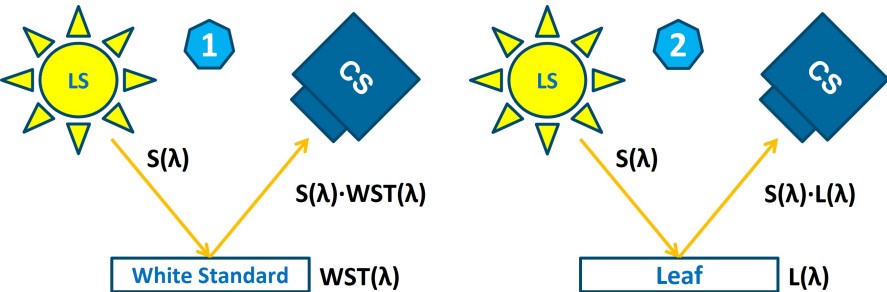

**Figure 6.** Measurement system and principle of spectral reflectance.

Next (Figure 6, right), the white standard was replaced by a leaf (spectral reflectance $L(\lambda)$). The spectral radiance $S(\lambda) \cdot L(\lambda)$ was measured by the same spectroradiometer *CS*. The light source spectrum (Equation (7)) and the spectral reflectance of the leaf were determined by Equation (8).

$$L(\lambda) = \frac{S(\lambda) \cdot L(\lambda) \text{ measured by CS}}{S(\lambda) \text{ determined previously in Equation (7)}} \tag{8}$$

*2.9. Plant Parameters Indicated by Reflectance Data*

Mathematical description: to build a model, Equations (9)–(14) are introduced. The investigated parameters—*TPC*, *TFC*, *ChA*, *ChB*, *CaT* and *SLW*—are called generally parameter *X*.

The spectral reflectance is represented by a matrix ($27 \times 401$) with the matrix elements ($R_{i,j}$, $j$ = 380–780, $i$ = 1–27 ). This matrix modified by a weighting function was also called weighting vector with the elements of $W_{380}$ to $W_{780}$.

The matrix multiplication of the spectral reflectance matrix and its weighting vector results in the intermediate vectors with the elements ($X_{\text{interm.},k}$, $k$ = 1–27) shown in Equation (9). This intermediate vector was used for calculation of the prediction vector with linear equation as in Equation (10).

$$\begin{pmatrix} X_{\text{intern.},1} \\ \dots \\ X_{\text{intern.},27} \end{pmatrix} = \begin{pmatrix} R_{1,380} & \dots & R_{1,780} \\ \dots & \dots & \dots \\ R_{27,380} & \dots & R_{27,780} \end{pmatrix} \cdot \begin{pmatrix} W_{380} \\ \dots \\ W_{780} \end{pmatrix} \tag{9}$$

$$\begin{pmatrix} X_{\text{predic.},1} \\ \dots \\ X_{\text{predic.},27} \end{pmatrix} = a \cdot \begin{pmatrix} X_{\text{intern.},1} \\ \dots \\ X_{\text{intern.},27} \end{pmatrix} + b \tag{10}$$

For the next steps, the statistical parameters *ErrorVector*, sum of square error (*SSE*), the average values between the experimental values and their mean values (*AVR.*) and $R^2$ were calculated as in Equations (11)–(14).

$$ErrorVector = \begin{pmatrix} X_{\text{predic.},1} \\ \dots \\ X_{\text{predic.},27} \end{pmatrix} - \begin{pmatrix} X_{\text{exper.},1} \\ \dots \\ X_{\text{exper.},27} \end{pmatrix} \tag{11}$$

$$SSE = \sum_{i}^{27} ErrorVector_i^2 \tag{12}$$

$$AVR. = \sum_{i}^{27} \left( X_{\text{experi.},i} - \frac{\sum_{i}^{27} X_{\text{experi.},i}}{27} \right)^2 \tag{13}$$

$$R^2 = 1 - \frac{SSE}{AVR.} \tag{14}$$

**\* Optimisation algorithm**: Subsequently, the optimisation algorithm was applied to determine parameters of the model. The optimisation algorithm is described schematically in Figure 7. In the experiments, the plants were cultivated under different conditions to create a comprehensive experimental database for constructing the prediction models. Specifically, the *TPC*, *TFC*, *ChA*, *ChB*, *CaT* and *SLW* of the cultivated plants were measured, processed and determined to create the biological database (denoted Database 1 in Figure 7). The spectra (Section 2.8) constitute Database 2 (Figure 7). The processing of Databases 2 with Equations (9)–(14) resulted in the statistical values sum of square error (*SSE*) and $R^2$. Both the weighting elements of the weighting vector ($W_1-W_{780}$ in Equation (9)) and linearity parameters (*a* and *b* in Equation (10)) were generated by the optimisation algorithm fminsearch (Matlab Optimisation Toolbox [41,42]). It represents a nonlinear unconstrained multivariable optimisation mission, where the variables are

the weighting function elements and the mathematical correlations are Equations (9)–(14). The optimisation loop was implemented automatically and continuously until *SSE* was minimal and $R^2$ was maximal. The achieved results represent the weighting vectors and linearity parameters of the model.

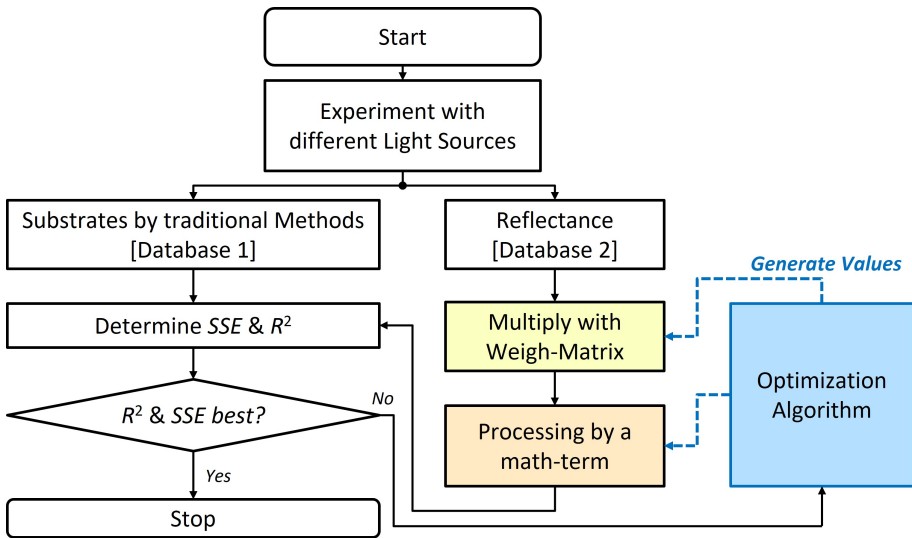

**Figure 7.** The schematic description of the synthesis optimisation algorithm for prediction models.

## 3. Results

### 3.1. Analysis of Biological and Biochemical Parameters (Database 1)

We analysed six different parameters, $TPC, TFC, ChA, ChB, CaT$ and $SLW$, from 27 plants, which were cultivated under three different light conditions, white, red, and blue/red light; and harvested on three different days, day 5 ($D_5$), day 14 ($D_{14}$) and day 21 ($D_{21}$) after the onset of the experiment (Figure 8).

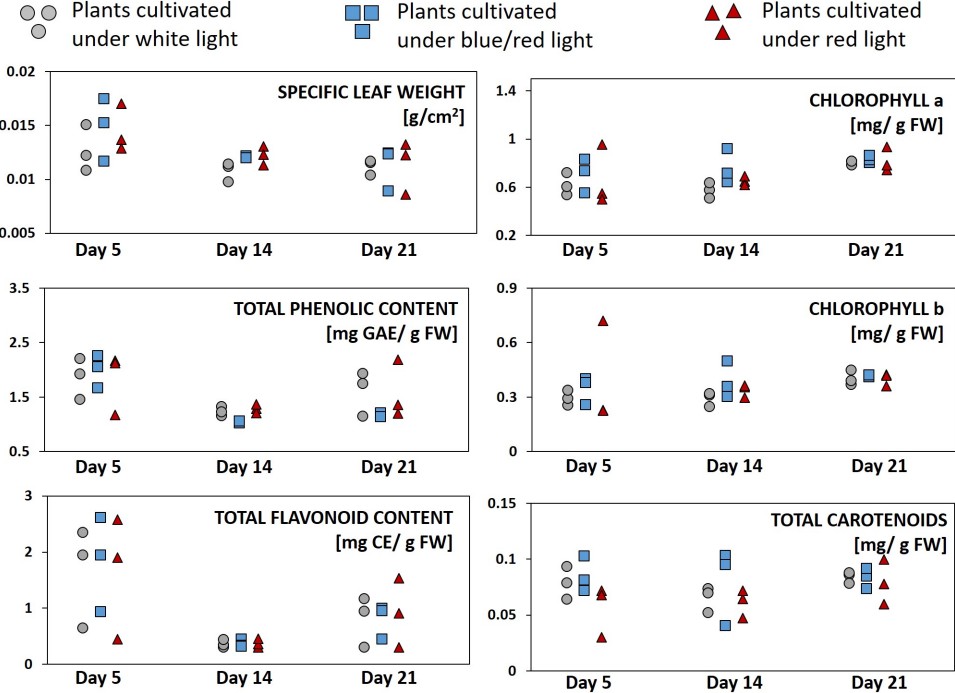

**Figure 8.** Analysis of six different plant parameters. Each symbol represents an individual plant. $GAE$—gallic acid equivalent; $CE$—catechin equivalent; $FW$—fresh weight.

Our results indicate significant variations of the investigated plants, even in those from the same groups (i.e., cultivated under the same light conditions and harvested on the same days). We performed the Kruskal–Wallis test to compare between plant groups categorised by either growth light conditions (white ($W$), red ($R$) or blue/red ($B/R$)) or by age (i.e., the harvesting day, 5 ($D_5$), 14 ($D_{14}$) or 21 ($D_{21}$)). Our results (Table 1) indicate that there was no statistically significant difference ($p > 0.05$) for different light conditions. However, the age of the plants modified five out of the six investigated parameters significantly ($p < 0.05$).

**Table 1.** Kruskal–Wallis test performed on plants grouped by growth light conditions (white ($W$), red ($R$) and blue/red ($B/R$)) or by age (harvested on day 5 ($D_5$), 14 ($D_{14}$) or 21 ($D_{21}$)). *SLW*—specific leaf weight; *TPC*—total phenolic content; *TFC*—total flavonoid content; *ChA*, *ChB*, *CaT*: chlorophyll a, chlorophyll b and total carotenoids contents.

| | KRUSKAL-WALLIS TEST | | | | | | |
|---|---|---|---|---|---|---|---|
| | **Light Condition Groups ($W, R, B/R$)** | | | | **Age Groups ($D_5, D_{14}$ & $D_{21}$)** | | |
| | $\chi^2$ | $df$ | $p$ | | $\chi^2$ | $df$ | $p$ |
| *SLW* | 4.361 | 2 | 0.113 | *SLW* | 6.62 | 2 | 0.037 |
| *TPC* | 2.384 | 2 | 0.304 | *TPC* | 10.31 | 2 | 0.006 |
| *TFC* | 0.575 | 2 | 0.75 | *TFC* | 13.04 | 2 | 0.001 |
| *ChA* | 2.931 | 2 | 0.231 | *ChA* | 8.6 | 2 | 0.014 |
| *ChB* | 2.215 | 2 | 0.33 | *ChB* | 8.7 | 2 | 0.013 |
| *CaT* | 5.527 | 2 | 0.063 | *CaT* | 2.93 | 2 | 0.231 |

### 3.2. Measurements of Plant Reflectance (Database 2)

The spectral reflectance of leaves growing under different light conditions (red, blue/red and white) was measured at different time points (5, 14 or 21 days after the start of the experiment). Each plant is represented by 10 different point measurements, which were taken with 10 randomly excised leaf discs. All spectra share the same common pattern with a left slope at about 525 nm and a flatter right slope at about 600 nm, which meet at the maximal point at about 550 nm. Furthermore, a concave point at about 690 nm with an amplitude of about 10% for all leaves was observed. The region above 700 nm is characterised by a steep slope, which appears at about 710 nm and levels off from 720 nm to a plateau with an amplitude of about 60%. We also observed minor differences among the 10 measured points (Figure 9). Such differences were highest after 5 days and then decreased. After 21 days, spectral reflectance was nearly constant.

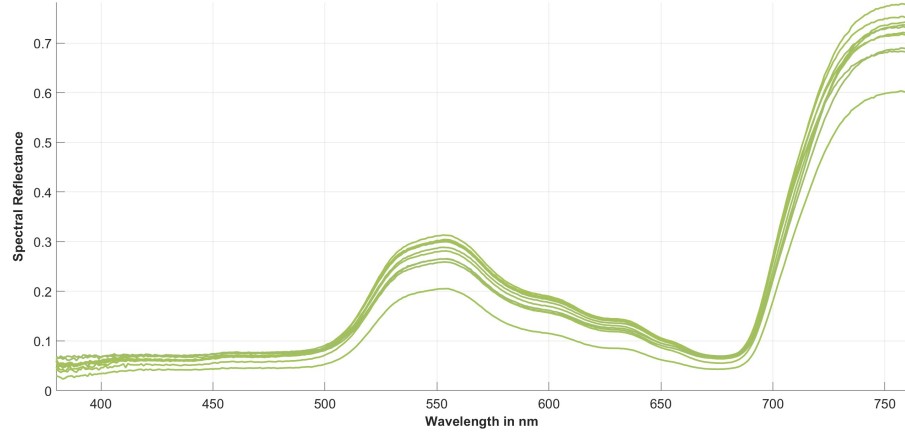

**Figure 9.** Leaf spectral reflectance. Displayed are 10 measurement runs for plant 1, grown under blue/red light and harvested on day 14.

### 3.3. Building the Prediction Models

The six investigated parameters—*TPC*, *TFC*, *ChA*, *ChB*, *CaT* and *SLW*—were optimised and synthesised with the described optimisation algorithm (Table 2). It can be concluded that only for *TPC* and *TFC*, good correlations between experimental values and the prediction models based on the optimised weighting functions exist. In the four remaining cases of *SLW*, *ChA*, *ChB* and *CaT*, the correlations with the experimental values are weak ($R^2 = 0.209$ with *SLW*, $R^2 = 0.198$ with *ChA* and $R^2 = 0.151$ with *ChB*) or very weak ($R^2 = 0.021$ with *CaT*).

**Table 2.** Optimisation results of all parameter cases.

| Parameter | $R^2$ | *SSE* | *dfe* | *Adj.R²* | *RMSE* | $a_{eval.}$ | $b_{eval.}$ |
|---|---|---|---|---|---|---|---|
| Total Phenolic Content (*TPC*) | 0.962 | 0.179 | 24 | 0.96 | 0.086 | 0.998 | 0.0034 |
| Total Flavonoid Content (*TFC*) | 0.876 | 1.886 | 25 | 0.88 | 0.274 | 1.004 | 0.0067 |
| Specific Leaf Weight (*SLW*) | 0.209 | $8.8 \times 10^{-5}$ | 25 | 0.178 | 0.0019 | 0.998 | 0.209 |
| Chlorophyll a (*ChA*) | 0.151 | 0.45 | 25 | 0.12 | 0.13 | 7.296 | 4.516 |
| Chlorophyll b (*ChB*) | 0.198 | 0.211 | 25 | 0.166 | 0.092 | 1.061 | 0.028 |
| Total Carotenoids (*CaT*) | 0.021 | 0.064 | 25 | −0.019 | 0.019 | 0.055 | 0.071 |

### 3.4. Correlations between Reflectance and TPC/TLC

\* **Total phenolic content (***TPC***)**: Optimisation results of total phenolic content (*TPC*) are shown in detail in Figure 10. The upper subplot depicting the comparison between the experimentally determined and predicted *TPC* shows a significant linear correlation. Statistical parameters $R^2$, *dfe*, *Adj.-R²* and *RMSE* are 0.96179, 24, 0.9602 and 0.086236, respectively. Linearity parameters—$a_{eval.} = 0.9972$ and $b_{eval.} = 0.0034217$—also compare favourably with the ideal ones, $a_{eval.} = 1$ and $b_{eval.} = 0$. The residual plot (middle subplot) shows that all absolute residuals are under 0.2.

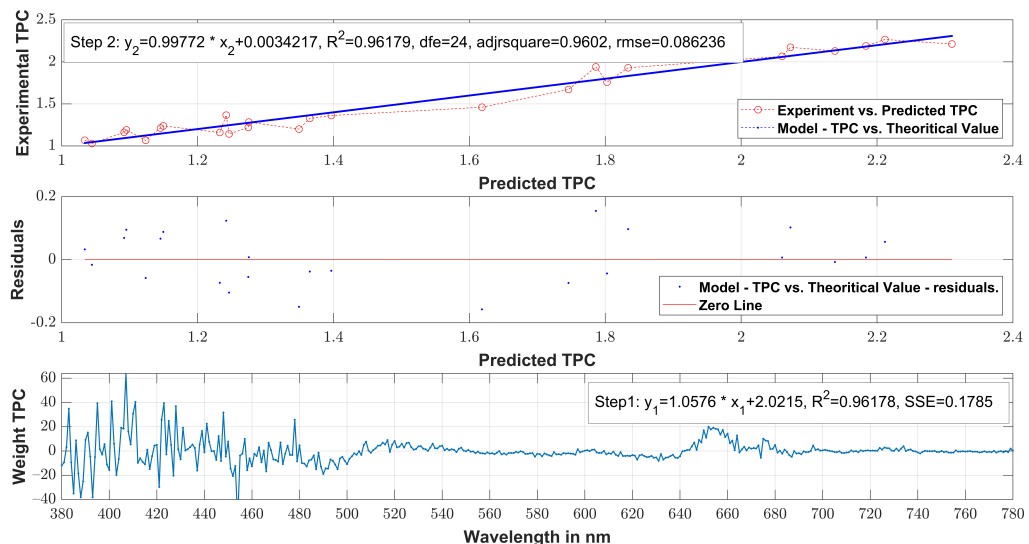

**Figure 10.** Prediction model, absolute residuals and weighting function plots of total phenolic content (*TPC*).

The weighting function is of importance for the optimisation algorithm. After optimising the weighting function, remaining parameters of the Equations (9)–(14) could be checked. The weighting function is displayed in the lower subplot. Here, the important wavelength ranges occur in the short wavelength range of from 380 to 500 nm and in the long wavelength range between 640 and 680 nm, which differs from the ranges of between 500 and 640 nm and from 680 to 780 nm. These remain low and constant.

**\* Total flavonoid content**: A comparable situation occurred in the case of *TFC* (Figure 11). The statistical parameters of the *TFC* model are somewhat inferior to those of the *TPC*, but still very good, with $R^2 = 0.87546$ and $SSE = 1.8862$. Absolute values are also larger. In the *TFC* weighting function, wavelength intensities stay relatively low and constant in regions between 520–640 nm and between 700 and 780 nm. Like in the *TPC* model, the weighting function for *TFC* is dominated by two wavelength ranges of 380–520 nm and 640–700 nm. However, the patterns in these ranges are more complex.

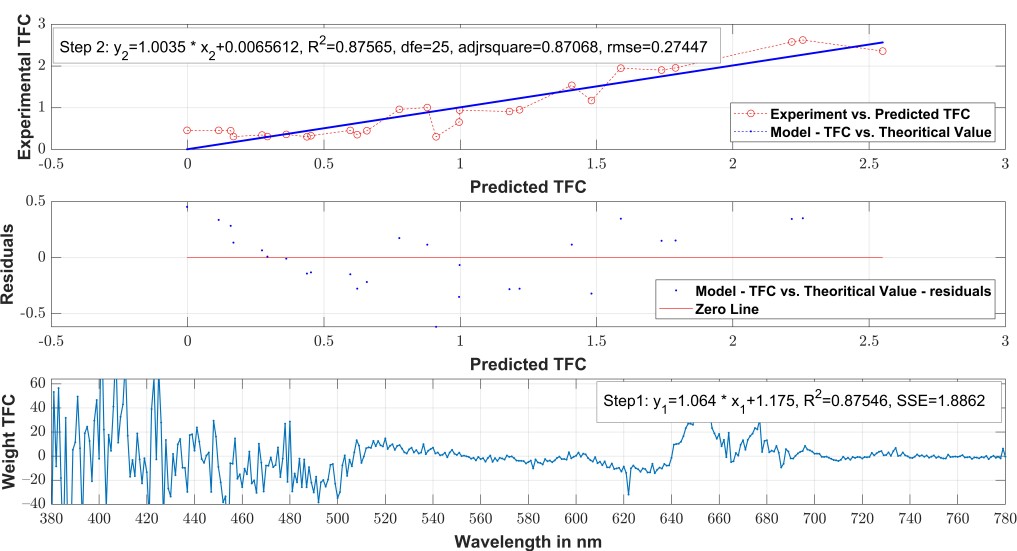

**Figure 11.** Prediction model, absolute residuals and weighting function plots of total flavonoid content (*TFC*).

## 4. Discussion

Accurate predictions of various biological and biochemical parameters of basil based on its reflectance properties have been achieved elsewhere [43,44]. However, the practical applicability of these findings is limited by the high cost of reflectance measuring equipment. As stated above, we applied several restrictions to our experiments to simulate the actual limitations of colour sensors, including the mode of acquisition (point measurements instead of hyperspectral imaging) and wavelength range (visible region only). It is significant that even under these restrictions, we were able to establish good correlations between reflectance data and *TPC* or *TFC*. These two parameters represent the contents of phenolic substances or flavonoids—two classes of secondary metabolites with important roles in the health-promoting effects and in the taste and flavour of sweet basil. The results imply that the use of spectral sensors for quantification of *TPC*/*TFC* promises a reasonable probability of success and should be further investigated.

Both multi-channel spectral sensors and low-cost sensors can be utilised for such applications. The former, as demonstrated by the examples reported in [27–31], can process many wavelength channels simultaneously. Using the established weighting functions (Figures 10 and 11), the measured reflectance from these channels can be converted by simple mathematic transformations into the predicted *TPC*/*TFC* values. On other hand, low-cost sensors often have only one or a couple of photodiodes and some simple built-in electronic elements. They need a different analytical approach. We propose that in this case, the weighting functions should be physically processed by additional optical filters which are specifically tailored to match the features of the weighting functions. In the regions where wavelength intensities stay relatively low and constant (500–640 nm and 680–780 nm in the case of *TPC* and 520–640 nm and 700–780 nm in the case of *TFC*), simple optical filters will be sufficient. On the other hand, for the dominant regions (380–500 nm and 640–680 nm in the case of *TPC* and 380–520 nm and 640–700 nm in the case of *TFC*), sophisticated filter designs will be required.

The study confirmed the applicability and scope of visible spectral sensors for analysis and quantification of important plant properties—in particular, the contents of valuable substances, such as phenolic compounds and flavonoids (corresponding to Technical Readiness Level 3, *TRL*3). Further work will be required to shift this concept into higher *TRL* levels. We also need to overcome other technical challenges that were left unaddressed in this study, such as the reliance of reflectance measurements to measuring distance and direction, and the interference of reflectance from neighbouring leaves. The prediction models should also be built and validated with a larger dataset of plants from a wider range of cultivars and growth conditions, and using more advanced machine learning methods. Regardless, the establishment of the TPC and TFC weighting functions is a significant step forward. Only with these weighting functions could the successive processing techniques of spectral sensors be realised for the envisaged fully automated and accurately monitored/controlled vertical farming systems.

**Author Contributions:** Conceptualisation: T.N.T., V.Q.T., K.Q.T. and R.K. (Ralf Kaldenhoff); performance of experiments: T.N.T., R.K. (Rieke Keller) and V.Q.T.; data analysis: T.N.T. and V.Q.T.; writing—original draft preparation, T.N.T. and V.Q.T.; writing—review and editing, T.N.T., V.Q.T., K.Q.T. and R.K. (Ralf Kaldenhoff). All authors have read and agreed to the published version of the manuscript.

**Funding:** This work received no specific grant from any funding agency in the public, commercial or not-for-profit sectors. The publication of the manuscript was supported by the Open Access Publishing Fund of the Technical University of Darmstadt.

**Institutional Review Board Statement:** Not applicable.

**Informed Consent Statement:** Not applicable.

**Data Availability Statement:** All data generated or analysed to support the findings of the present study are included this article. The raw data can be obtained from the authors, upon reasonable request.

**Conflicts of Interest:** The authors declare no conflict of interest.

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
