# Peer review of "Multi-Channel Spectral Sensors as Plant Reflectance Measuring Devices—Toward the Usability of Spectral Sensors for Phenotyping of Sweet Basil (Ocimum basilicum)"

_agronomy, doi:10.3390/agronomy12051174_

Round 1
Reviewer 1 Report
Manuscript deals with the application of optoelectronic devices in modern agriculture to control state of the plants. Authors have grown 27 plants and they propose a model for the phenolic content on this group of plants. The manuscript is divided into typical paragraphs - the structure is fine. However, I find some lacks.
The title of the manuscript and the introduction suggest experiment with the multispectral detector as sensing system. The literature review looks fine and gives enough introduction to the problem, but later in the experiment a spot spectroradiometer was used and there are no validating experiments with the multispectral sensors. From this point of view the content is something different than presented in the title and abstract. Therefore, the title of the manuscript does not reflect its content.
Considering the experiment procedure:
- it is not clear which of standard geometries was used for the measurements of spectral reflectance of the leaves. Figures 3 and 6 are not clear in this point.
- Was the experiment statistically pre-analyzed? Is this number of plants enough? What was the calculated power of the test? The results mentioned on page 9 in lines 218/219 are surprising – no difference for various spectra of light? In my opinion your experiment was not correctly planned – not enough plants and no repeatability of spectral measurements. You also suggest that in the discussion paragraph.
- Equation 1 – I suppose it should be multiplied by 4 – d was a diameter, not radius of the disk
- Description of the “measurement system and spectral features” is not clear. What is light source in this description? What source have you used for these spectral measurements? What is “spectral production”? How did you measure spectral reflectance – in what geometry? How did you assure repeatability of measurement conditions? For spectral reflectance measurement integrating spheres are usually used.
- Figure 6 is incorrect – on the right you should replace “white standard” with the “leaf disc” and probably different name instead of WST
Page 3 - lines 120-122 are repeated starting from line 124 on page 3 and later on page 4
Author Response
Dear Sir/Madam as Reviewer,
Many thanks for your very valuable comments, questions, and corrections!
We have studied much from your interesting visions and ideas. You helped us much for improving our manuscript. In the attached PDF-file, we have answered your points in the form of dialogs between reviewers and authors.
Thank you again and best regards,
Vinh Trinh - On behalf of all co-authors

Reviewer 2 Report
The authors demonstrated the practicality of color sensors for plant phenotyping (specifically sweet basil) in an indoor growing condition. Although the experimental setup and collection of reflectance spectra at different conditions were sound, I am not convinced with the modeling part. Here are my observations:
- The authors described the modeling part in Section 2.9: Plant parameter indicated by reflectance data. The process looks very similar to a feed-forward neural network. However, the optimization procedure was not explained well. The authors mentioned “The optimization loop is implemented automatically and continuously until SSE is minimal and R2 is maximal”. But how did they do that? For example, in neural network, gradient descent or stochastic gradient descent or Adam (which is very popular now-a-days) are used as optimization algorithm. In the optimization, there is also the concept of learning rate. Learning rate indicates how fast the weights can be updated. Opposed to these already established methods, the implemented method in this manuscript is too naïve and unclear. It is also confusing about the use of both SSE and R2 as the metrics of the optimization.
- Even if we assume that the optimization algorithm and the mathematical model is sound, the performance should be compared with traditional approaches. For example, simple linear regression, partial least squares regression (which is a very powerful yet simple regression for reflectance-level phenotyping), support vector regression or random forest regression. Finally, since the proposed approach is similar to feed-forward neural networks, it should be compared with that as well.
- The R2, SSE, dfe, and so on listed in Table 2 are also problematic. The values were generated from the entire set of dataset. Since the proposed modeling approach is a supervised learning method, the performance of the model has to be evaluated for a separate set of dataset. That’s why often researchers randomly divide the dataset into training and testing set, where the testing set is kept separate from the training/learning stage. The performance evaluation should only be done with the testing set.
Author Response

(The authors gave the same response as above.)

Reviewer 3 Report
The study predicted various basil properties using reflectance information measured by a spectroradiometer. The manuscript can be and needs to be improved by clearly defining the scope, research question, and objectives of the study. A statement such as line 307-308 is too general for a scientific publication. A more comprehensive review of similar studies can also much improve the manuscript quality. Below are some of my observations.
- Line 90-107, maybe briefly discuss CCD and CMOS (I later realized the study used a spectroradiometer rather than regular RGB cameras, so maybe have a paragraph explaining spectroradiometer mechanism)
- Line 96, it would be a good idea to also mention other color-separation mechanisms in RGB cameras.
- Line 102, maybe explain what “Clear” and “Flicker” mean. They are uncommon terms.
- Line 120-122, redundant
- Goals and objectives are not explicitly specified in Introduction. It is unclear what research problems the study is trying to solve.
- Literature review of existing studies that have done similar research work is missing in Introduction. Also the knowledge gap in current literature is not specified. This is a major weakness of the manuscript.
- Line 152, why these three light conditions were used?
- Figure 4, typo “Absolut”
- Add manufacturer’s information for CS2000
- Is CS2000 a camera or a spectroradiometer? Double check wording.
- Figure 6, remove the redundant one. Also add a real photo of using the camera take measurements.
- Line 193-205, explicitly explain what the optimization algorithm is and how it optimizes model parameters.
- What is the purpose of Figure 9? I do not see a big difference between the subfigures.
Author Response

(The authors gave the same response as above.)

Round 2
Reviewer 1 Report
Respected Authors,
Thank you for some improvements and explanations. I am pretty sure the content is ok considering methodology of analysis (although the number of samples is still very small - for example p values are much over 0,05 for some results which may suggest the number of cases was very far for expected). But unfortunately I still disagree with some fudamental issues.
I still don't like some obvious terms (e.g. "spectral production"), but this is not of fundamental importance. In this case I suggest to use widely accepted terminology. In my opinion "spectral production" is spectral radiance or spectral luminance of the white standard / leaf sample area when illuminated by the reference source. Please, clarify that in the manuscript.
Then on the basis of these measurements you have calculated reflectance. However both - luminance and radiance - are strongly dependent on the direction and this is why I disagree with the correctness of the measurement procedure. According to my experience direction of illumination and measurement is very important in such experiments if instruments with limited FOV are used (I performed some very similar experiments with my master student several years before). The basil leafs are not diffusive. In case of diffusive object that problem could be neglected, but in this case it may include random error in recorded spectrum. Of course that can be controlled till some extent, but you haven't presented that in your manuscript. Please, clarify at least in the text how you assured constant conditions during experiment and clearly state the limitations of your research.
Could you please add more analysis (at least theoretical, maybe with some graphics to clarify some thoughts) on the aplicability of spectral sensors? This is of main interest in your research but the discussion of your results in this context seems rather poor.
Author Response
Dear Sir/Madam Reviewer,
Many thanks for your very good corrections! We greatly appreciate your inputs and have improved our manuscripts accordantly
Thank you again and best regards,
Vinh Trinh - On behalf of all co-authors

Reviewer 2 Report
I thank the authors for reviewing the suggestions carefully. However, I still believe that the evaluation metrics (R2, RMSE, etc.) should be calculated for a "test set" which has not been used while training the model. Additionally, training set evaluation metrics can also be mentioned. A 70/30 train/test split can be done. In this way, we can see if the optimization model is overfitting or not.
Author Response

(The authors gave the same response as above.)

Reviewer 3 Report
The revised manuscript title is too general to be an article title. It sounds rather like a book title. The “instead of hyperspectral cameras” part is unnecessary. I recommend the authors just focusing on basil.
Line 162-166, the research questions are not well-written. The first question is too general. Also there must be substantial existing studies that predicted plant biological/ biochemical characteristics using reflectance data, which should be comprehensively reviewed in Introduction. In the second question, “spectral weighting functions” is rather abrupt since there is no context regarding this term in Introduction. The third question make no sense to me.
Regarding my previous comments “Line 152, why these three light conditions were used?” and “Line 193-205, explicitly explain what the optimization algorithm is and how it optimizes model parameters.”, please incorporate your responses into the manuscript.
Author Response
Dear Sir/Madam Reviewer,
Many thanks for your very good corrections! We greatly appreciate your inputs and have improved our manuscripts accordantly
Thank you again and best regards,
Vinh Trinh - On behalf of all co-authors

This manuscript is a resubmission of an earlier submission. The following is a list of the peer review reports and author responses from that submission.